# Leveraging a synthetic biology approach to enhance BCG-mediated expansion of Vγ9Vδ2 T cells

Christine M. Qabar[1], Allison W. Roberts[2], Lucas M. Waldburger[3,4],
Edward E. K. Baidoo[5,6], Emine Akyuz Turumtay[5,6], Jay D. Keasling[3,5,6,7], Dan A. Portnoy[1,2],
Jeffery S. Cox[2]*

1 Department of Plant and Microbial Biology, University of California Berkeley, Berkeley California,
United States of America, 2 Department of Molecular and Cell Biology, University of California Berkeley,
Berkeley California, United States of America, 3 Department of Bioengineering, University of California
Berkeley, Berkeley California, United States of America, 4 Department of Electrical Engineering and
Computer Science, University of California Berkeley, Berkeley California, United States of America,
5 Joint BioEnergy Institute, Lawrence Berkeley National Laboratory, Emeryville California, United States
of America, 6 Biological Systems and Engineering Division, Lawrence Berkeley National Laboratory,
Berkeley, California, United States of America, 7 Department of Chemical and Biomolecular Engineering,
University of California Berkeley, Berkeley California, United States of America

* jeff.cox@berkeley.edu

## Abstract

There is an urgent need to develop a more efficacious anti-tuberculosis vaccine as the current live-attenuated vaccine strain BCG fails to prevent pulmonary infection in adults. In this study, we leverage a synthetic biology approach to engineer BCG to produce more (E)-4-hydroxy-3-methyl-but-2-enyl pyrophosphate (HMBPP), an intermediate of bacterial—but not host—isoprenoid biosynthesis via the methyleryth-ritol phosphate (MEP) pathway. HMBPP strongly activates and expands Vγ9Vδ2 T cells, which are unique to higher-order primates and protect against *Mycobacterium tuberculosis* infection. BCG has been engineered to produce specific ligands and antigens to some success; in contrast, our strategy exploits a self-nonself recognition mechanism in the host via HMBPP sensing, which has not been attempted before. To inform the design of our recombinant strains, we performed synteny analyses of >63 mycobacterial species and found that isoprenoid biosynthetic genes are not operonic across all the 356 surveyed genomes, but some genes are frequently found in pairs. Thus, we generated synthetic loci with the goal of specifically overproducing HMBPP and tested the ability of these engineered strains to induce human Vγ9Vδ2 expansion in an *in vitro* stimulation assay. We found that BCG expressing a synthetic MEP locus significantly enhanced Vγ9Vδ2 T cell expansion over the wild-type vaccine strain, and overexpression of the HMBPP synthase GcpE alone potently induced Vγ9Vδ2 T cell expansion with no downregulation of other pathway genes. Together these engineered strains present two successful strategies to accumulate HMBPP and overcome feedback inhibition of the MEP pathway.

**Data availability statement:** All relevant data are within the paper and its Supporting information files.

**Funding:** This work was supported by the National Institute of General Medical Sciences (T32GM132022 to CMQ), the National Institute of Allergy and Infectious Diseases (U19AI162583 and U19AI135990 JSC), (1P01 AI063302 and 1R01AI027655 to DAP), as well as the National Science Foundation Graduate Research Fellowship (DGE-1752814 to LMW) and the Henry Wheeler Center for Emerging and Neglected Diseases Irving H. Wiesenfeld Fellowship (CNCND-32305-47950 to CMQ). This material was based upon work supported by the Joint BioEnergy Institute, U.S. Department of Energy, Office of Science, Biological and Environmental Research Program with Lawrence Berkeley National Laboratory (DE-AC02-05CH11231 to JDK). Funders did not play any role in the study design, data collection and analysis, decision to publish, or preparation of the manuscript. There was no additional external funding received for this study.

**Competing interests:** I have read the journal's policy and the authors of this manuscript have the following competing interests: J.D.K. has financial interests in Ansa Biotechnologies, Apertor Pharma, Berkeley Yeast, BioMia, Cyklos Materials, Demetrix, Lygos, Napigen, ResVita Bio, and Zero Acre Farms. D.A.P. has a financial interest in Laguna Biotherapeutics. The other authors declare no competing interests. This does not alter our adherence to PLOS ONE policies on sharing data and materials.

## Introduction

*Mycobacterium tuberculosis* (*Mtb*), the causative agent of tuberculosis disease, has been persistent throughout the course of human history. Most notably in the late 1800s to early 1900s, *Mtb* infected nearly the whole population of Europe and resulted in ~25% mortality [1]. Despite the discovery of the bacterium over 142 years ago, *Mtb* remains a global health threat, killing approximately six thousand individuals per day and latently infecting an estimated quarter of the world's population [2].

There is an urgent need to develop new vaccines with greater efficacy than the current Bacille Calmette-Guérin (BCG) vaccine, a live-attenuated strain of *Mycobacterium bovis* lacking several *Mtb*-specific virulence genes [3,4]. BCG has been used globally since 1921 despite its failure to prevent pulmonary infection in adults [5,6]. Several recombinant BCG (rBCG) vaccine candidates are currently in the clinical pipeline, many challenges exist toward the development of an improved vaccine, both biological (correlates of protection, epitope selection, limited animal models) and epidemiological (ethics of human trials, exclusion of pregnant women) in nature [1].

Building on the existing BCG vaccine platform is appealing due to its established safety profile in humans, ease of production and transport, and relatively inexpensive cost at a median $0.24 per dose [7]. Other attempts to engineer BCG focus on heterologous expression of antigens and immune ligands, modification and encapsulation of surface molecules, and induction of bacterial lysis to varying degrees of protection [8–12]. In contrast, our strategy exploits a specific metabolic sensing pathway in the host. Our approach is to enhance the immunogenicity of BCG by modulating the production of an endogenously synthesized immunostimulatory molecule in the bacterium. Specifically, we focused on activating Vγ9Vδ2 T cells, a small but important subset of T cells that are unique to humans and non-human primates [13]. Vγ9Vδ2 T cells constitute the most abundant subset of γδ T cells in humans, comprising 60−95% of all circulating γδ T cells and have been reported to expand to up to 50% of total blood T cells upon infection with pathogens including *Mtb* [14–16]. These T cells respond specifically and potently to a unique diphosphate molecule called (E)-4-hydroxy-3-methyl-but-2-enyl pyrophosphate (HMBPP), which is produced by most bacteria as an intermediate of isoprenoid biosynthesis [17–21].

Isoprenoids represent a large, important class of biomolecules found in all domains of life, with as many as 95,000 isoprenoid natural products identified to date including cholesterol, heme, and vitamin K [22,23]. In bacteria, isoprenoids support fundamental cellular processes, including membrane fluidity, cell wall synthesis, electron transport, stress response, and signaling and communication [24,25]. Isoprenoids are essential in *Mtb*, which relies on the isoprenoid decaprenyl phosphate for the synthesis of the essential cell wall components lipoarabinomannan and the mycolyl-arabinogalactan-peptidoglycan complex, as well as the "linker" molecules between arabinogalactan and peptidoglycan in the cell wall [26–28]. Further, *Mtb* employs the isoprenoid virulence factors tuberculosinol and isotuberculosinol to arrest phagosomal maturation during infection [29–31]. Together, isoprenoids are critical to all domains of life and represent essential molecules for *Mtb* growth and pathogenesis.

Despite wide structural and functional diversity, all isoprenoids arise from the fundamental precursor isopentenyl pyrophosphate (IPP), which can be synthesized via two independent pathways: the mevalonate (MEV) pathway, which is primarily found in eukaryotes, archaea, and some bacteria, or the methylerythritol phosphate (MEP) pathway, which is found in bacteria, plants, and algae [32,33] (Fig 1A). Importantly, HMBPP is only produced via the MEP pathway, which is essential in *Mtb* [34].

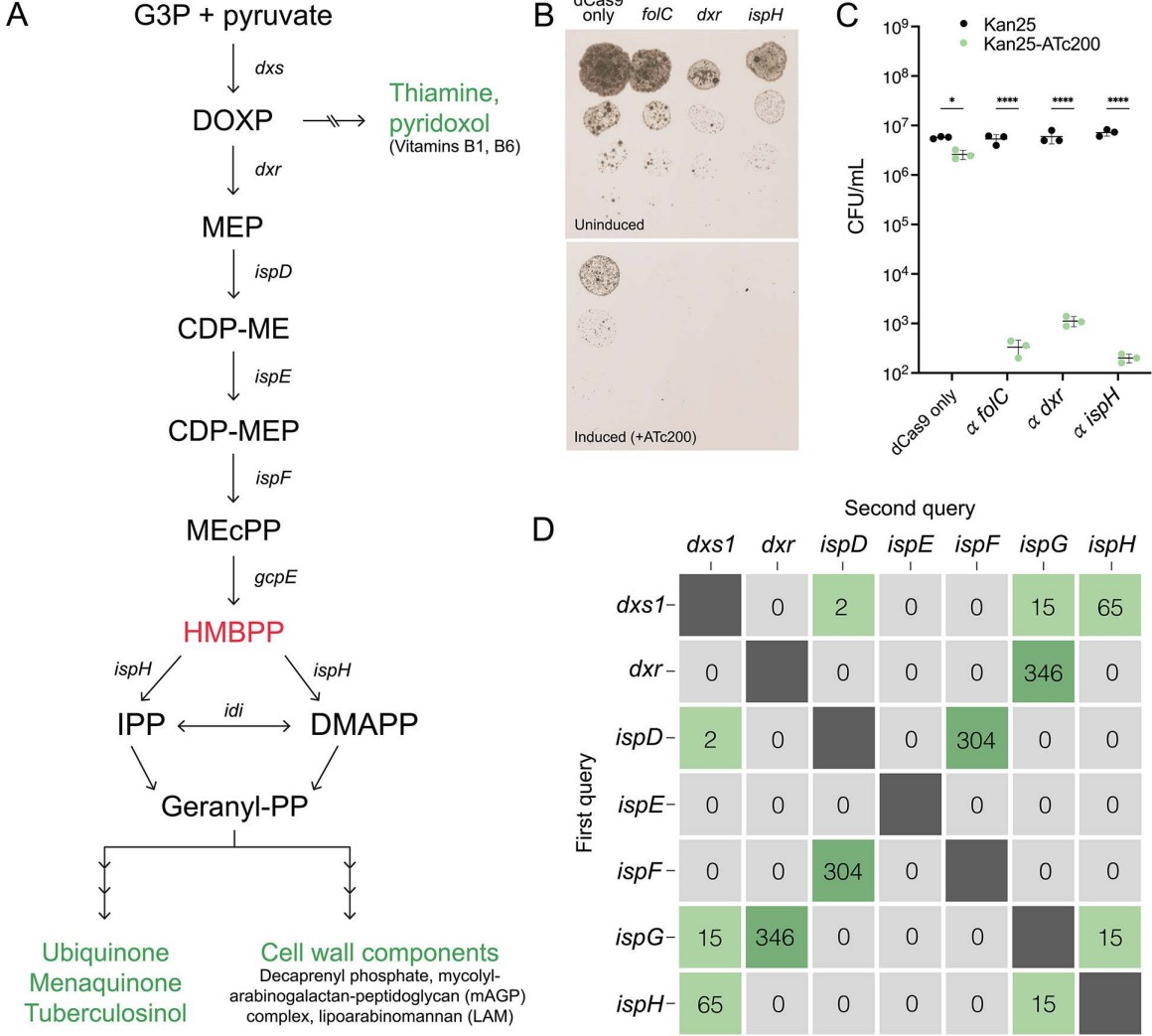

**Fig 1. The MEP pathway is essential in BCG. A. The MEP pathway of isoprenoid biosynthesis.** Shown in green are end products and shunts; shown in red is HMBPP, the activator of Vγ9Vδ2 T cells. Genes encoding enzymatic steps are as follows: *dxs*, DOXP synthase; *dxr*, DOXP reductoisomerase; *ispD*, MEP cytidylyltransferase; *ispE*, CDP-ME kinase; *ispF*, MEcPP synthase; *gcpE*, HMBPP synthase; *ispH*, HMBPP reductase; *idi*, isopentenyl diphosphate isomerase. **B. Silencing of MEP genes is lethal in BCG.** Targeting of *dxr* and *ispH* by inducible CRISPR interference was lethal in BCG. Target genes, including *folC* and a dCas9 only plasmid as controls, were transcriptionally repressed by the addition ATc **(B)**. CFU without (Kan25) and with (Kan25-ATc200) induction were counted and analyzed via two-way ANOVA followed by Sidak's multiple comparisons (B; *p ≤ 0.05, ****p < 0.0001). Targeting of *folC, dxr*, or *ispH* caused a significant decrease in viability, while the dCas9-only control experienced notably less death (C; *p ≤ 0.05, **p ≤ 0.01, ***p < 0.001, one sample t test). Shown is mean ± SD. N = 3 biological replicates. **C. Synteny analysis reveals naturally occurring gene pair biases to leverage in our synthetic platform.** Several gene pairs are frequently found in proximity to each other across mycobacteria: *dxs1* + *ispH*; *dxr* + *ispG/gcpE*; *ispD* + *ispF*; and occasionally *ispH* + *ispG/gcpE*. Synteny analysis was performed on 353 mycobacterial genomes using each gene as a query to generate this 7x7 matrix of co-occurrence.

The Vγ9Vδ2 T cell receptor (TCR) indirectly senses bacterial HMBPP via an "inside-out" mechanism by which HMBPP binds the cytosolic domain of the transmembrane butyrophilin BTN3A1 in an antigen presenting cell [35–39]. Upon HMBPP binding, the surface-associated BTN3A1:BTN2A1 heterodimer undergoes a conformational change which is recognized by the Vγ9Vδ2 TCR [14,38]. These noncanonical T cells also respond to endogenous host IPP [40], however HMBPP is 10,000-fold more potent an activator than IPP [41], demonstrating a considerable bias toward detecting microbial-derived metabolites. Together, the strength and specificity of this interaction suggests that the Vγ9Vδ2-phosphoantigen sensing pathway is likely predominantly a self-nonself detection mechanism.

Once activated, Vγ9Vδ2 T cells produce important antimicrobial factors including granulysin, granzyme A, perforin, tumor necrosis factor alpha (TNF-α), granulocyte-macrophage colony-stimulating factor (GM-CSF), and interferon gamma (IFNγ) [42–47]. Importantly, IFNγ in turn leads to the production of reactive nitrogen species, which can control bacterial growth *in vivo* [48]. Further, HMBPP-stimulated Vγ9Vδ2 T cells support the canonical αβ T cell response to infection by producing IL-12, the major cytokine driving Th1 polarization of T cells which is crucial to a protective anti-*Mtb* response [44,49]. Finally, activated Vγ9Vδ2 T cells have prolific memory recall abilities, expanding 60-fold in HMBPP-immunized non-human primates [44] and a reported memory sustained up to seven months post exposure [50]. Importantly, activation of Vγ9Vδ2 T cells is protective *in vivo*. Studies in non-human primates have demonstrated that treatment with an HMBPP analog, immunization with HMBPP-producing *Listeria monocytogenes*, or adoptive transfer of *ex vivo* activated Vγ9Vδ2 T cells protects against *Mtb* infection [44,45,51]. Thus, HMBPP-mediated activation and subsequent expansion of Vγ9Vδ2 T cells is an attractive target for improving the BCG vaccine.

In this study, we leveraged a synthetic biology approach to metabolically engineer BCG to produce excess HMBPP and thus, stimulate a more robust Vγ9Vδ2 T cell response. A similar approach of expressing a synthetic MEP operon has been previously used to generate a "microbial cell factory" for economically-relevant isoprenoid production in *Bacillus subtilis* [52]. However, in this study we leverage this approach to improve the immunogenicity of the BCG vaccine. We tested our rBCG strains in primary human cells and found that both a single-gene overexpression platform and a synthetic MEP locus significantly expanded human Vγ9Vδ2 T cells. Together, these data provide critical *in vitro* validation of metabolically reengineered BCG and can be built upon to develop a more efficacious anti-*Mtb* vaccine.

## Materials and methods

### Bacterial strains and culture

*Mycobacterium tuberculosis* variant *bovis* BCG Pasteur (ATCC 35734) was routinely grown in Middlebrook 7H9 liquid medium or 7H10 agar (Difco) supplemented with 10% OADC (oleic acid-albumin-dextrose-catalase) and 0.05% Tween80. *E. coli* strains DH5α or NEB® 10-beta were grown in LB broth or agar and used for propagation and cloning of plasmids. When required, the following antibiotics were used: kanamycin (25 μg/ml for mycobacteria, 50 μg/ml for *E. coli*); hygromycin (50 μg/ml for mycobacteria, 150 μg/ml for *E. coli*); zeocin (25 μg/ml for mycobacteria, 50 μg/ml for *E. coli*); anhydrotetracycline (200 ng/ml). Institutional Biosafety approval was obtained for all biological agents and genetic manipulations under BUA #407 at the University of California, Berkeley.

### Molecular cloning

Primers, oligos, and plasmids used in this study are listed in S1 Table in S1 File. Standard electroporation protocols were used for the transformation of plasmids into *E. coli* and mycobacteria. 50uL of electrocompetent DH5α *E. coli* was combined with 5ul of ligation product or plasmid DNA and incubated on ice for 30 min, then heat shocked at 42°C for 30 sec, chilled on ice for 5 min, and recovered at 37°C in Luria broth for one hour. Cells were then plated on selective LB agar and incubated 37°C overnight. Electrocompetent mycobacteria were prepared by washing mid-log (OD 0.5–0.8) cultures four times in decreasing volumes of sterile 10% glycerol. The final resuspension concentrates cultures ~20-25x, and electrocompetent cells can be used fresh or frozen at −80°C. 200ul of electrocompetent cells were combined with 5ul of plasmid

DNA in a 0.2 cm electroporation cuvette, and cells were electroporated with a single pulse at 2.5 kV with 25 μF capacitance and 1,000 Ω resistance. Transformants were recovered overnight at 37°C in 7H9 media. The following day, cells were plated on selective 7H10 agar containing the appropriate antibiotic. 3–6 clones were randomly selected and verified via PCR.

## CRISPRi

Gene silencing and gene deletion were performed using the site-specific transcriptional repression system CRISPRi as previously described [53]. Briefly, guide oligos were annealed and ligated into the integrative, dCas9-containing pLJR965 vector and the subsequent plasmids were transformed into BCG. A dCas9-only vector plasmid was also included as a control in the transformation and induction assay. Inducible transcriptional repression was achieved by treatment with 200 ng/ml anhydrotetracycline (ATc). Bacterial input was $OD_{600}$-normalized to 0.4, spots were plated on 7H10 agar supplemented either with kanamycin or kanamycin and ATc, and all plates were incubated for 12 days prior to imaging and CFU counting.

## Synteny analysis

To identify MEP synteny across mycobacterial genomes, we used the program 'core analysis of syntenic orthologs to prioritize natural product gene clusters' (CORASON) [54] with some updates. Briefly, 353 *Mycobacterium* genomes representing at least 62 unique species were downloaded from the NCBI Genome database [55]. A database of protein sequences was generated using DIAMOND v4.0.515 [56] and queried with the IspE protein sequence from *Mycobacterium bovis* BCG. Results were filtered to include 5 protein sequences upstream and downstream of the IspE match. We created a second DIAMOND database of protein sequences within the neighborhood of IspE and queried with the remaining MEP protein sequences from BCG. The number of MEP genes within the gene neighborhood of IspE across the *Mycobacterium* genomes were counted and are represented in Fig 1D. To query whether MEP genes are operonic in any bacterium, we downloaded 36,193 bacterial reference genomes from NCBI and searched for all MEP genes in close genomic proximity. A new DIAMOND database was generated from the reference genomes and queried for the IspE protein sequence as previously described. Gene neighborhoods were searched for matches to the remaining MEP pathway genes.

## Generation of synthetic constructs

### (I) Expression of a synthetic MEP locus (pCQ88)

The pMV306.hyg backbone was digested with KpnI and XbaI to generate non-compatible ends. The genes *dxs1/BCG_2695, dxr/BCG_2892, ispD/BCG_3647, ispE/BCG_1068, ispF/BCG_3646,* and *gcpE/BCG_2890* were included; HMBPP reductase *ispH* was not included to optimize HMBPP accumulation. The six genes were further subdivided into two-gene operons, driven by unique strong promoters and preceding individual terminators, to optimize expression of each gene (Fig 2A). To ensure all genes are expressed, genes were assembled into two-gene operons informed by the genomic pair biases identified in our bioinformatic analysis (Fig 1D).

As *dxr* and *gcpE* are found within an operon in BCG, the genes were placed in an operon driven by PgroEL. Unique RBSs were placed upstream of each gene; the RBS from *groEL* upstream of dxr and the native RBS upstream of *gcpE*. 30 bp downstream of the *gcpE* stop codon were included to capture the native terminator. The genes *ispD* and *ispF* were placed in an operon as they are operonic in BCG, with overlapping coding regions. The strong mycobacterial promoter Psmyc and its associated RBS was placed upstream of ispD with no intergenic RBS due to the ORF overlap. 30 bp downstream of the *ispF* stop codon were included to capture the native terminator. The genes *dxs1* and *ispE* were placed under control of the MOP promoter from pMH406. The Hops RBS was placed upstream of *dxs1* and the *gcpE* RBS was placed upstream of *ispE*. The rrnB T1 terminator from pMV261 was placed downstream of *ispE* to strongly terminate expression of the end of the three operons.

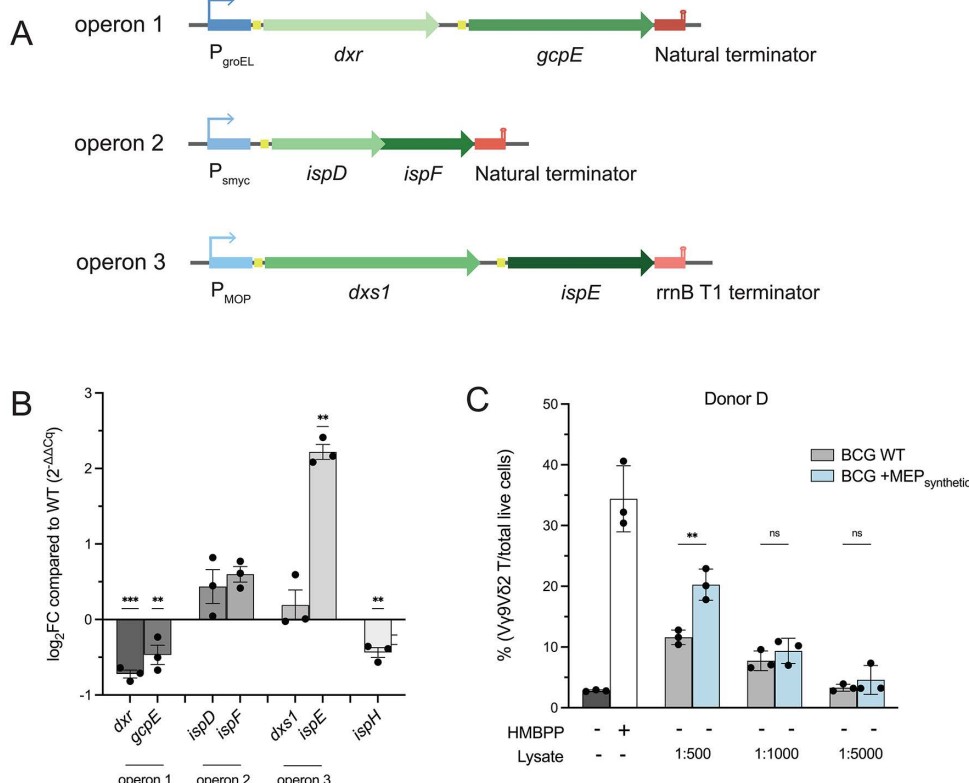

**Fig 2. Generation of a synthetic MEP locus. A. Designing pCQ88, a synthetic MEP locus.** Using gene pairs identified in our synteny analysis, we designed an integrative plasmid carrying an engineered biosynthetic gene cluster comprising all upper-branch MEP genes. Each operon is composed of one pair of genes (green) driven by a strong, unique mycobacterial promoter (blue), preceded by a ribosome binding site (yellow), and followed by either an exogenous or natural terminator (red). **B. Genes in the synthetic MEP locus are variably expressed.** Operon 1 (*dxr, gcpE*) and the endogenous *ispH* were significantly downregulated, while operons 2 (*ispD, ispF*) and 3 (*dxs1, ispE*) were moderately upregulated compared to WT (One sample t test; **p ≤ 0.01, ***p ≤ 0.001). Expression data for *ispH* is included as a control as it was not encoded in the synthetic MEP synthetic locus. Shown is mean ± SD. N = 3 biological replicates in technical triplicate. **C. The engineered biosynthetic gene cluster MEP**<sub>synthetic</sub> **significantly enhances Vγ9Vδ2 T cell expansion over baseline.** There is a significant increase in expansion induced by BCG + MEP$_{synthetic}$ (Sidak's multiple comparisons; **p ≤ 0.01). Treatment with purified HMBPP (0.1 ng/mL) is included as a control. Shown is mean ± SD from one independent donor in technical triplicate, representative of three independent donors (data from other donors are found in S1 Fig).

Resulting operons were optimized to achieve a GC content of <65% while preserving codon usage. Final, codon-optimized operons were ordered from Twist Bioscience (San Francisco, CA), amplified from the delivery vector, and cloned into a final vector via Gibson assembly. Transformants were confirmed via whole plasmid sequencing (Plasmidasaurus, San Francisco, CA).

### (II) Overexpression of HMBPP synthase GcpE (pCQ105)

The pMV306.hyg backbone was digested as described above. The strong mycobacterial promoter Psmyc from the plasmid pUV15 was PCR amplified using primers oCQ404 and oCQ405 to introduce 5' homology to the KpnI end of the digested vector and 3' homology to the *gcpE* insert. The gene *gcpE* was PCR amplified from BCG gDNA using primers oCQ406 and oCQ407 to introduce 5' homology to the Psmyc insert and 3' homology to the XbaI end of the digested vector. All homologous overhangs are 20 bp. The two inserts were assembled into the digested vector via Gibson assembly and transformed into NEB® 10-beta competent *E. coli*. Transformants were confirmed via whole plasmid sequencing.

## RNA isolation and RT-PCR

Cultures were grown to mid-log ($OD_{600}$ ~ 1.0), pelleted, resuspended in 1 ml Trizol, and lysed via bead beating thrice in 30 second increments, incubating on ice for five minutes in between bursts. RNA was then extracted from the supernatant via chloroform-ethanol extraction, washed using a PureLink RNA Mini kit (Invitrogen), and treated with DNAse I. Following a 10-minute DNAse heat inactivation at 75°C, RNA was stored at −80°C. Primers were designed with the following criteria: 17−25 bp in length, 3' G/C clamp, 75−150 bp amplicon, Tm ~ 60ºC. Primers were also analyzed for self- and cross-secondary structure. cDNA was prepared using a SuperScript III First-Strand Synthesis Kit (Invitrogen). Briefly, random hexamer primers were annealed to template RNA, cDNA was synthesized using reverse transcriptase, and residual RNA was digested by RNAse H. cDNA was stored at −20°C. The SsoAdvanced Universal SYBR Green Supermix (BioRad) was used for RT-PCR reactions on genes of interest as well as 16S for normalization. cDNA samples were run in technical triplicate on a CFX Connect Real-Time System (BioRad) running CFX Manager software. ΔΔCq values were $log_2$ transformed and presented as $log_2$ fold change over WT [57].

## PBMC expansion assays

Expansion assays were performed as previously described [15,34]. Mid-log cultures were OD-normalized to $OD_{600}$ = 0.5 and low-molecular-weight filtrates were generated by washing 15 ml in DPBS and lysing via bead-beating in Lysing Matrix B tubes (MP Bio). Lysates were then pelleted and the supernatant fractionated using 3kDa filter cartridges (Amicon). Lysate supernatant fractions <3 kDa were used for subsequent assays as this fraction contains microbial HMBPP. Equivalent protein input across lysate inputs was confirmed via BCA assay. Primary human peripheral blood mononuclear cells (PBMCs) were acquired from and deidentified by the Stanford Blood Center (Stanford, California, USA) in accordance with the University of California, Berkeley Human Research Protection Program's written policy on Research with Coded Private Information or Biological Specimens. Cells were cultured in R10 media (RPMI-1640 with 2mM glutamine, 10mM HEPES, 10% FBS, 50 U/ml pen-strep, and 50µM β-Mercaptoethanol) supplemented with 100 U/ml IL-2. PMBCs were plated at a density of $7.5 \times 10^5$ cells/well in a U-bottom 96-well plate and incubated overnight at 37°C, 5% CO2, with humidity. The following day, PBMCs were stimulated with either pure HMBPP or bacterial lysates at various dilutions. Fresh media was added on day 4, and cells were harvested, stained with antibodies, and fixed on day 6. Flow cytometry was performed on day 7. Antibody panel is as follows: Vδ2-PacBlue (1:50, clone B6, Biolegend cat #331414), Vγ9-APC (1:50, clone B3, Biolegend cat #331310), CD3-PerCP-Cy5.5 (1:100, clone OKT3, Biolegend cat #317337), αβ TCR-FITC (1:16.67, clone IP26, Biolegend cat #306706).

## Metabolite extraction + metabolomic analysis

The metabolite extraction protocol was modified from previously described mycobacterial extraction [58]. Briefly, cells were grown to $OD_{600}$ 0.8–0.9 and the equivalent of 10 $OD_{600}$ units were pelleted, resuspended in metabolite extraction buffer (2:2:1 methanol:acetonitrile:water), and lysed via bead beating six times in 30 second increments at 4°C. Lysate was then pelleted, and supernatant was fractionated through 3 kDa ultra centrifugal filter columns (Amicon, catalog #UFC500324). Extracts were held at −80°C until targeted metabolomic analysis. Liquid chromatography–mass spectrometry (LC-MS) analysis of metabolites was performed as previously described [59].

## Statistical analysis

For CFU-based assays, data were analyzed via two-way ANOVA followed by Tukey's multiple comparisons test. To calculate $log_2$ fold change ($log_2$FC) in viability, a ratio of induced to input CFU was calculated and $log_2$ transformed, and a one sample t test was used to determine whether the $log_2$FC was significantly different from a theoretical mean of 0, indicating no change in viability. For RT-PCR data, ΔΔCq values were calculated and $log_2$-transformed and a one sample t test was

used to determine whether the $\log_2$FC of each gene was significantly different from a theoretical mean of 0, indicating no difference in expression compared to WT. For PMBC expansion assays and metabolite measurements, data were analyzed via ordinary one- or two-way ANOVA followed by Tukey's or Sidak's multiple comparisons test, respectively, depending on the number of conditions in the comparison.

## Results

### The MEP pathway is essential in BCG

Most mycobacteria solely utilize the MEP pathway for isoprenoid biosynthesis (Fig 1), with the exception of a *M. marinum-* derived clade [60–62]. Further, multiple transposon mutagenesis and gene deletion experiments have identified all MEP genes as essential in *Mtb* [34,63–68], and because BCG and *Mtb* are closely related, we predicted that the MEP pathway is also essential in BCG. Indeed, we were unable to obtain a chromosomal deletion of *dxr* or *ispH*, two genes in the MEP pathway. To definitively demonstrate that these genes are essential, we turned to CRISPR interference (CRISPRi) to inducibly repress these genes in BCG. Using single guide RNAs that target *dxr* or *ispH*, as well as against the known essential gene *folC*, we found that repression of either gene inhibited growth and significantly reduced viability (Fig 1B, C), supporting that the MEP pathway is essential in BCG. Together, these data represent the first evidence to demonstrate the essentiality of the MEP pathway in BCG, presenting barriers to future engineering of isoprenoid biosynthesis.

### Synteny analysis of MEP genes across mycobacterial genomes

Given the evidence supporting a protective role for Vγ9Vδ2 T cells in *Mtb* infection, we hypothesized that enhanced Vγ9Vδ2 activation might also enhance protection. To engineer a strain of BCG that produced elevated levels of HMBPP, we initially set out to combine all MEP genes leading to HMBPP synthesis into a single locus to maximize expression. To rationally design this synthetic MEP locus, we first sought to determine whether any bacterial species already encode all MEP genes either in proximity or in an operon. Most bacteria do not encode the MEP pathway as an operon [69], which suggests that in all bacteria—and more saliently, in all mycobacteria—the MEP genes are distributed across the genome. To determine whether any MEP genes are found together in mycobacteria, we queried 353 mycobacterial genomes for co-occurrences of MEP genes clustered in genome neighborhoods. The synteny analysis revealed several gene pairs that are frequently found in proximity: *dxs1+ispH; dxr+ispG/gcpE; ispD+ispF*; and occasionally *ispH+ispG/gcpE* (Fig 1D). Because these pair biases are strongly represented across the genus, we sought to preserve them in our subsequent engineering attempts.

### Generation of a synthetic MEP locus

Our synteny analysis findings posed a unique challenge: if all MEP genes in all bacteria are scattered across the genome, we must carefully design a synthetic locus which takes into account the tendency for certain gene pairs to be found in proximity to one another, as well as the potential that each gene in the MEP pathway may be regulated differently. Using the pair biases revealed in our synteny analysis, we designed the plasmid pCQ88, an integrative plasmid carrying three, two-gene operons composed of strong promoters, defined ribosome binding sites, and strong terminators to prevent read-through transcription (MEP$_{synthetic}$; Fig 2A). We transformed this plasmid expressing the six genes *dxr, gcpE, ispD, ispF, dxs1, ispE* into BCG (BCG+MEP$_{synthetic}$) and confirmed successful integration via junction PCR. Importantly, engineering this strain did not result in a change in biomass or CFU per OD compared to wild-type BCG (S3 Fig).

To determine if this strain was more stimulatory to Vγ9Vδ2 T cells, we used an *in vitro* expansion assay using peripheral blood mononuclear cells (PBMCs) from human donors. Animal models were not at our disposal because Vγ9Vδ2 T cells are restricted to higher order primates [70]. In our *in vitro* expansion assay, primary human PBMCs were treated *ex vivo* with lysates from either WT BCG or BCG+MEP$_{synthetic}$ and assessed for Vγ9Vδ2 expansion via flow cytometry. We found that BCG+MEP$_{synthetic}$, our first iteration of a synthetic BCG strain, significantly enhanced Vγ9Vδ2 expansion over WT baseline

(Figs 2C and S1). qRT-PCR analysis revealed that only some of the ectopically expressed genes were in fact overexpressed over WT levels (Fig 2B). Interestingly, we observed downregulation of *dxr* and *gcpE*, two critical steps in the isoprenoid biosynthetic pathway (Fig 2B), although this clearly did not impact the ability of this strain to expand Vγ9Vδ2 T cells.

One limitation to our synthetic platform of MEP pathway expression is the potential for feedback inhibition by IPP, DMAPP, and other downstream metabolites. Intracellular levels of MEP intermediates are tightly regulated via negative feedback mechanisms [71,72]. Thus, it is possible that ectopic expression of the synthetic locus overwhelmed the cell with metabolites such that the cell throttled all IPP synthesis. Indeed, we observed downregulation of *ispH*, which was not ectopically expressed in the synthetic locus, suggesting that even the native pathway may have been experiencing feedback inhibition (Fig 2B). Together, these challenges prompted us toward a new engineering strategy to maximize Vγ9Vδ2 T cell stimulation by BCG.

## Generation of a *gcpE* single gene overexpression platform

To avoid burdening the cell by overexpressing multiple metabolic genes, we turned to a simpler approach in which only one gene is ectopically expressed. We attempted to overexpress *dxr* as it is the first committed step in the pathway; however, our initial attempts to transform a *dxr*-overexpressing plasmid into BCG failed, whether *dxr* was under the control of a strong promoter or its native promoter. We were able to obtain transformants when *dxr* was placed under an anhydrotetracycline-inducible promoter but observed no growth upon induction, suggesting that overexpression of *dxr* alone is toxic in BCG, which has been reported previously in *Mtb* [34]. This is likely because accumulation of early MEP intermediates is toxic. We instead expressed the HMBPP synthase GcpE to specifically accumulate HMBPP, as it likely will not accumulate other intermediates in the pathway (BCG+$gcpE_{OE}$; Fig 3A). A similar strategy was successfully engineered in *Mtb* [34], providing strong evidence that this approach would improve Vγ9Vδ2 T cell expansion in our system. We confirmed overexpression of *gcpE* and found no global pathway repression, in contrast to our previous construct (Fig 3B). Interestingly, we observed overexpression of the kinase IspE in our engineered strain in both constructs (Figs 2B and 3B), suggesting that expression of this gene may be responsive to a yet-undetermined stimulus.

We transformed the plasmid pCQ105 carrying this construct into BCG and the resultant strain had similar biomass and CFU per OD compared to WT (S3 Fig). Using our *ex vivo* Vγ9Vδ2 T cell stimulation assay, we tested this strain and found that it enhanced activation of Vγ9Vδ2 T cells compared to WT (Figs 3C and S2). This is in agreement with findings in *Mtb*, in which *gcpE* overexpression strongly enhanced activation over WT *Mtb* [34]. Together, these data suggest that a targeted overexpression strategy may circumvent the tight regulation of this pathway in BCG. Importantly, it is critical to the success of BCG engineering to determine how to overcome broader feedback inhibition to maximize HMBPP production, and thus vaccine immunogenicity, in this system.

## Metabolic profiling reveals minimal energetic and redox perturbations in engineered strains

To test whether our engineering functionally translated to enhanced isoprenoid precursor synthesis, we performed metabolic profiling of our strains compared to WT BCG. While cellular HMBPP occurred below the limit of detection [73], we found that both engineered strains had significantly higher levels of 2-C-methylerythritol 4-phosphate (MEP), the first committed intermediate of this pathway, as well as moderately enhanced levels of the intermediate 1-deoxy-D-xylulose 5-phosphate (DOXP; Fig 4A). There were no major differences in NAD(H) or NADP(H) redox couples among the strains (Fig 4B, C), and our measured ratios are in agreement with those reported for *Mtb* [74–79]. Interestingly, we did observe a notably lower NADH:NAD ratio in BCG+$MEP_{synthetic}$ in comparison to BCG (Fig 4B), indicating a more oxidative cellular environment that may prime cells to meet energetic demands [80], although this difference is not statistically significant. Further, there were no major differences in cellular concentrations of acetyl CoA, free CoA, ATP, or pyruvate, the starting reagent for the MEP pathway (Fig 4D). Together, metabolic profiling suggests our engineered strains had greater isoprenoid flux with minimal disruption of global cellular metabolism.

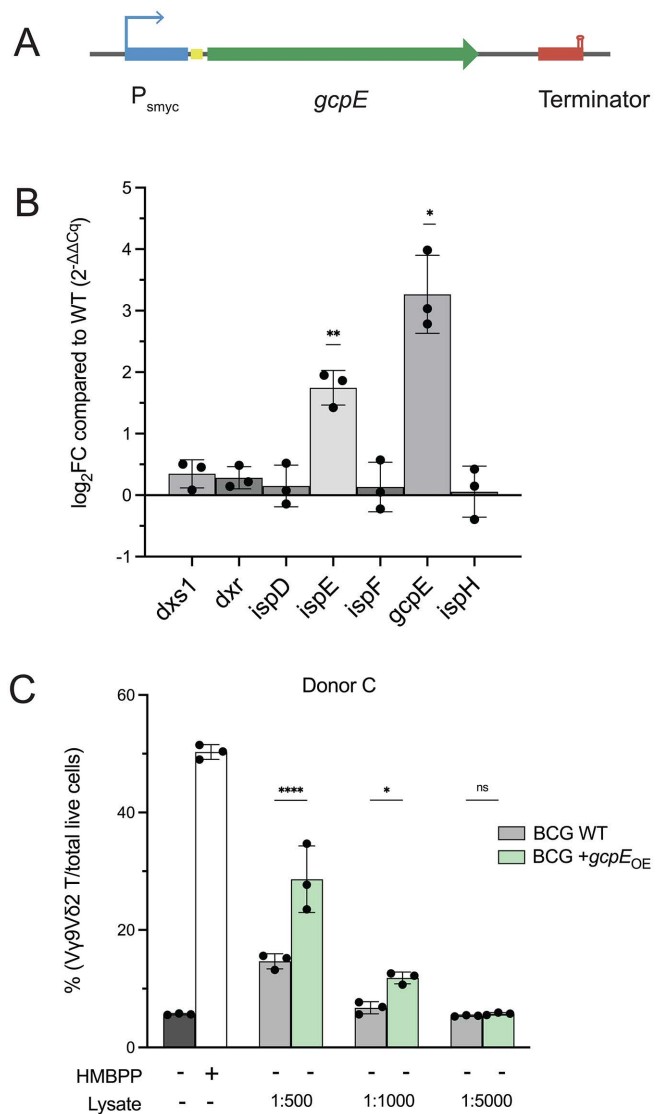

**A**

$P_{smyc}$   gcpE   Terminator

**B**

**C**

Donor C

BCG WT
BCG +gcpE_OE

| HMBPP | - | + | - | - | - | - | - | - |
| Lysate | - | - | 1:500 | 1:1000 | 1:5000 |

**Fig 3. Generation of a *gcpE* overexpression construct. A. Designing a *gcpE* OE construct.** The pCQ105 construct contains a single-gene expression platform composed of a strong mycobacterial promoter (blue) driving the expression of *gcpE* (green), which is preceded by a ribosome binding site (yellow) and followed by a terminator (red). **B. HMBPP synthase GcpE is strongly overexpressed.** Expression of each gene in the MEP pathway was compared between WT and pCQ105-expressing BCG. Compared to WT, *gcpE* was significantly overexpressed in the engineered strain (One sample t test; *p < 0.05, **p < 0.01). Shown is mean ± SD. N = 3 biological replicates in technical triplicate. **C. Overexpression of *gcpE* significantly enhances Vγ9Vδ2 T cell expansion over WT.** Shown is percent of Vγ9Vδ2 T cells when stimulated by either HMBPP or lysate from WT and BCG + pCQ105 cells, compared to baseline. There is a significant difference in Vγ9Vδ2 between WT and BCG + pCQ105 (Sidak's multiple comparisons; ***p < 0.001, *p < 0.05). Treatment with purified HMBPP (0.1 ng/mL) is included as a control. Shown is mean ± SD from one independent donor in technical triplicate, representative of four independent donors (data from other donors are found in S2 Fig).

## Discussion

HMBPP is an intermediate of isoprenoid metabolism via the MEP pathway and potently activates and expands host Vγ9Vδ2 T cells (Fig 1). We found that this metabolic pathway is essential in BCG (Fig 1B), as has been reported in *Mtb* [34,63,64]. Because Vγ9Vδ2 T cells are protective in the context of *Mtb* infection, we sought to generate a rBCG vaccine

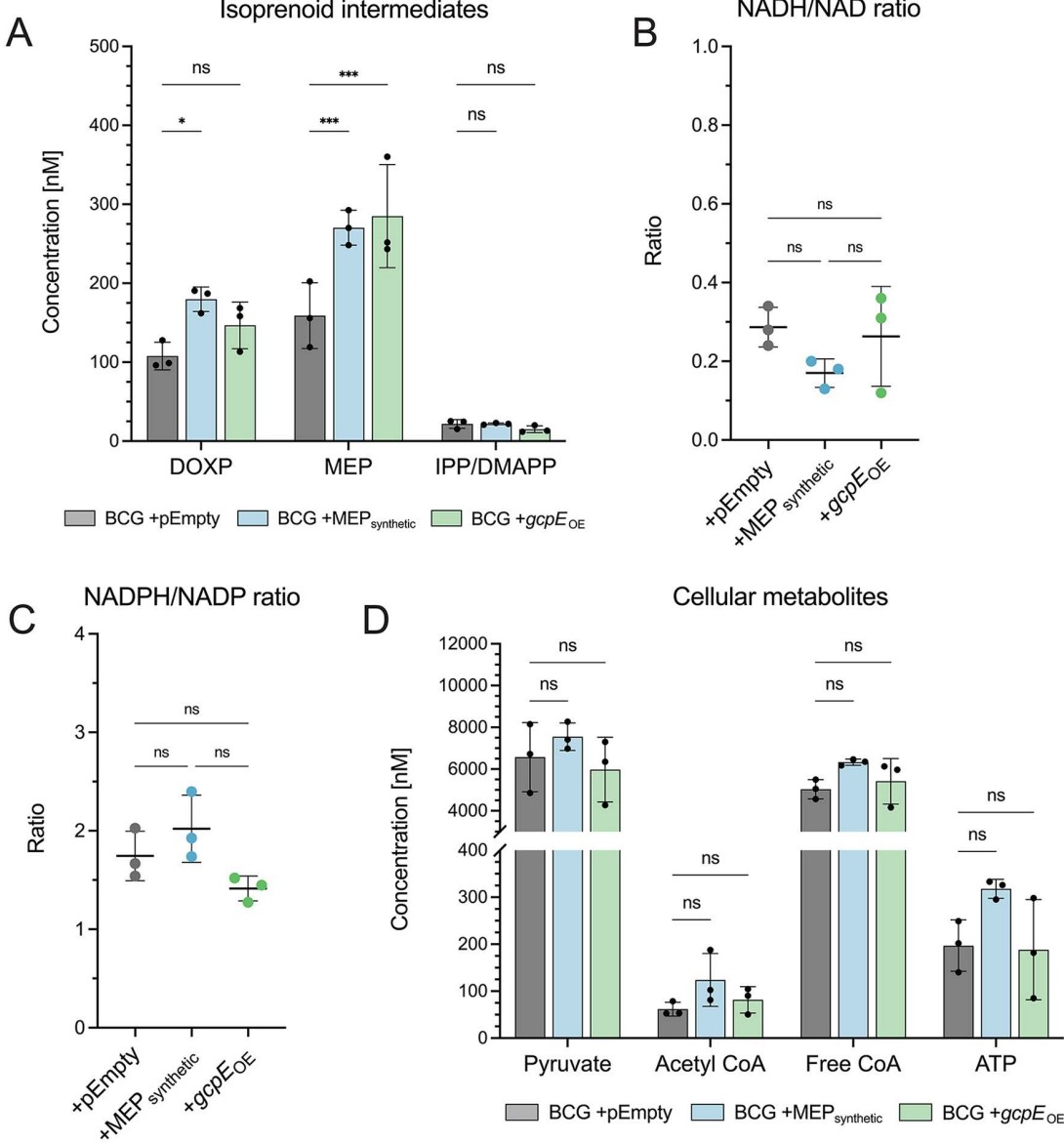

**Fig 4. Metabolic profiling of WT and engineered strains of BCG. A. Engineered strains of BCG accumulate more MEP intermediates than WT.** BCG+MEP<sub>synthetic</sub> and BCG+$gcpE_{OE}$ both have significantly more 2-C-methylerythritol 4-phosphate (MEP) than WT, and BCG+MEP<sub>synthetic</sub> had significantly more 1-deoxy-D-xylulose 5-phosphate (DOXP) compared to WT (Sidak's multiple comparisons; ***p<0.001, *p<0.05). Other MEP intermediates, including HMBPP, were not detected in our analysis. Shown is mean±SD. N=3 biological replicates per strain. **B, C. Cellular redox state is not impacted in the engineered strains.** There are no significant differences in NADH/NAD (**B**) and NADPH/NADP (**C**) ratios among the strains, although BCG+MEP<sub>synthetic</sub> has a moderately lower NADH/NAD ratio compared to WT. Shown is mean±SD. N=3 biological replicates per strain. **D. Major cellular metabolites are not impacted in the engineered strains.** There are no significant differences in intracellular acetyl CoA, free CoA, ATP, or pyruvate concentrations among the strains. Shown is mean±SD. N=3 biological replicates per strain.

strain that induced a stronger Vγ9Vδ2 response by synthesizing a bolus of HMBPP. Synteny analyses across mycobacterial genomes revealed certain pair biases of MEP genes, which we used to assemble a synthetic MEP locus as well as a single gene overexpression construct (Figs 2A and 3A). We found that the rBCG strain encoding the synthetic MEP locus did not have strong overexpression of MEP genes (Fig 2B), yet still significantly enhanced Vγ9Vδ2 T cell expansion

over wild type (Fig 2C). The *gcpE* overexpression strain also significantly expanded Vγ9Vδ2 T cells over WT BCG with no concomitant downregulation of other MEP genes (Figs 3B and 3C). Interestingly, we detected overexpression of the kinase IspE in both engineered strains (Figs 2B and 3B), presenting a potential role of this gene in the regulation of these pathways. Metabolomic analysis revealed that both engineered strains accumulated more MEP intermediates compared to WT BCG, with no apparent impairment of global cellular metabolism or redox state (Fig 4).

Other rBCG vaccines are in clinical trials, including the candidate VPM1002 which expresses the pore-forming hemolysin listeriolysin O from *L. monocytogenes* to activate autophagy during vaccination [1]. Our engineering strategy is distinct such that we modulated an endogenous biosynthetic pathway to accumulate a metabolic intermediate, which has not been attempted before in this system. An HMBPP-boosted strain of BCG has clinical relevance beyond tuberculosis; BCG is an FDA-approved treatment for bladder cancer immunotherapy and Vδ2 T cells are correlated with favorable outcomes during treatment [81]. However, challenges to modifying BCG remain, including the potential for any engineering perturbation to alter classical antigen production. Thus, understanding the molecular underpinnings of this phenomenon might reveal mechanisms by which we can manipulate the system to accumulate more HMBPP.

Future work is needed to develop a live infection model to determine whether the intracellular concentration of HMBPP in the engineered strains is relevant in the context of vaccination, as well as the mechanism by which HMBPP is released from phagocytosed bacteria into the cytosol of the macrophage. The bacterium may actively secrete HMBPP, although this is unlikely as a polar, charged molecule with no transporter identified to date, or it may be released through bacterial lysis in the phagosome. It remains to be determined whether HMBPP sensing requires access of the bacterium to the host cell cytosol via phagosomal perforation, or whether there is a yet-undetermined mechanism for HMBPP itself to cross the phagosomal membrane.

Finally, to optimize future engineering efforts using this strategy, it is critical to understand the mechanisms of isoprenoid feedback regulation and, importantly, how to overcome them. Additional improvements may be possible through the addition of D-glyceraldehyde-3-phosphate, which has been shown in other systems to strongly accelerate Dxs activity [82]. A complementary engineering strategy could be to mildly repress *ispH*, resulting in less HMBPP conversion and thus enhanced accumulation, although this would require careful modulation as full *ispH* repression is lethal (Fig 1B). Importantly, our engineered strains demonstrated significant Vγ9Vδ2 T cell expansion over baseline, suggesting protective immunity in the context of vaccination; however, validation of protective efficacy requires testing *in vivo*.

While much remains to be done on the path to developing a more protective vaccine against tuberculosis disease, we demonstrated the essentiality of the MEP pathway and explored promising potential avenues for metabolic reconstruction of BCG. Together, this study provides a novel investigation of isoprenoid biosynthesis pathway regulation in BCG, laying the groundwork to inform a rationally reengineered, Vγ9Vδ2-targeting vaccine strain.

## Supporting information

**S1 Fig. Supporting data for BCG expressing the synthetic MEP locus. A.** Shown are representative flow cytometry dot plots and gating strategy from donors stimulated with BCG + pCQ88. **B, C.** PMBC expansion assay data from two additional, independent donors stimulated with BCG + pCQ88 lysate. Data were analyzed via one-way ANOVA followed by Sidak's multiple comparisons; ***$p < 0.001$, **$p < 0.005$.
(EPS)

**S2 Fig. Supporting data for BCG overexpressing *gcpE*. A.** Shown are representative flow cytometry dot plots and gating strategy from donors stimulated with BCG + pCQ105. **B-D.** PMBC expansion assay data from two additional, independent donors stimulated with BCG + pCQ105 lysate. Data were analyzed via one-way ANOVA followed by Sidak's multiple comparisons; ****$p < 0.0001$, **$p < 0.005$, *$p < 0.05$.
(EPS)

**S3 Fig. Confirmation of accuracy of OD-normalization. A. Engineered strains do not have significantly different biomass than WT.** WT BCG and the strains expressing pCQ88 or pCQ105 were grown to late log phase, OD-matched to $OD_{600} = 1.0$ and weighed. Shown is mean ± SD. N = 3 biological replicates. **B. Engineered strains do not have significantly different OD:CFU ratios.** All strains were grown to late log phase, OD-matched to $OD_{600} = 0.5$ and plated on 7H10 agar. Colonies were counted after 14 days' incubation at 37°C. Data were analyzed via one-way ANOVA followed by Sidak's multiple comparisons test. Shown is mean ± SD. N = 3 biological replicates.
(EPS)

**S1 File. Tables S1 to S4.**
(XLSX)

## Acknowledgments

Many thanks to Preethi Thattai Ragunathan for her expertise and troubleshooting assistance, as well as Anne Xu, Miles Wingfield, Rebecca Procknow, and Zoila Álvarez-Aponte for learning alongside me. Thanks to Sarah Stanley, Russell Vance, Aziz Qabar, Astrid, and all members of the Cox lab for their helpful discussions and feedback.

## Author contributions

**Conceptualization:** Christine M. Qabar, Dan A. Portnoy, Jeffery S. Cox.

**Data curation:** Christine M. Qabar.

**Formal analysis:** Christine M. Qabar, Allison W. Roberts, Lucas M. Waldburger, Edward E. K. Baidoo, Emine Akyuz Turumtay, Jeffery S. Cox.

**Funding acquisition:** Jeffery S. Cox.

**Investigation:** Christine M. Qabar, Allison W. Roberts, Lucas M. Waldburger, Edward E. K. Baidoo, Emine Akyuz Turumtay, Jeffery S. Cox.

**Methodology:** Christine M. Qabar, Edward E. K. Baidoo, Emine Akyuz Turumtay, Jeffery S. Cox.

**Project administration:** Jeffery S. Cox.

**Resources:** Jay D. Keasling, Jeffery S. Cox.

**Software:** Lucas M. Waldburger.

**Supervision:** Jay D. Keasling, Dan A. Portnoy, Jeffery S. Cox.

**Validation:** Allison W. Roberts.

**Visualization:** Christine M. Qabar.

**Writing – original draft:** Christine M. Qabar.

**Writing – review & editing:** Christine M. Qabar, Jeffery S. Cox.

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
