## [Decision Letter · Decision Letter 0]

24 Jul 2025

Dear Prof. Cox,

Thank you for submitting your manuscript to PLOS ONE. After careful consideration, we feel that it has merit but does not fully meet PLOS ONE’s publication criteria as it currently stands. Therefore, we invite you to submit a revised version of the manuscript that addresses the points raised during the review process.

We look forward to receiving your revised manuscript.

Kind regards,

Atul Vashist, PhD

Academic Editor

PLOS ONE

Journal Requirements:

[This work was supported by the National Institutes of Health grants T32GM132022 (C.M.Q.), U19AI162583 (J.S.C.), U19AI135990 (J.S.C.), 1P01 AI063302 (D.A.P.), 1R01 AI027655 (D.A.P.), as well as the National Science Foundation Graduate Research Fellowship (L.M.W., https://nsfgrfp.org/) and the Henry Wheeler Center for Emerging and Neglected Diseases (C.M.Q., https://cend.globalhealth.berkeley.edu). Funders did not play any role in the study design, data collection and analysis, decision to publish, or preparation of the manuscript.].

[This work was supported by the National Institutes of Health grants T32GM132022 (C.M.Q.), U19AI162583 (J.S.C.), U19AI135990 (J.S.C.), 1P01 AI063302 (D.A.P.), 1R01 AI027655 (D.A.P.), as well as the National Science Foundation Graduate Research Fellowship (L.M.W., https://nsfgrfp.org/) and the Henry Wheeler Center for Emerging and Neglected Diseases (C.M.Q., https://cend.globalhealth.berkeley.edu). Funders did not play any role in the study design, data collection and analysis, decision to publish, or preparation of the manuscript.].

We note that one or more of the authors is affiliated with the funding organization, indicating the funder may have had some role in the design, data collection, analysis or preparation of your manuscript for publication; in other words, the funder played an indirect role through the participation of the co-authors. If the funding organization did not play a role in the study design, data collection and analysis,

decision to publish, or preparation of the manuscript and only provided financial support in the form of authors' salaries and/or research materials, please do the following:

1. Review your statements relating to the author contributions, and ensure you have specifically and accurately indicated the role(s) that these authors had in your study. These amendments should be made in the online form.

2. Confirm in your cover letter that you agree with the following statement, and we will change the online submission form on your behalf:

“The funder provided support in the form of salaries for authors [insert relevant initials], but did not have any additional role in the study design, data collection and analysis, decision to publish, or preparation of the manuscript. The specific roles of these authors are articulated in the ‘author contributions’ section.

[Many thanks to Preethi Thattai Ragunathan for her expertise and troubleshooting assistance, as well as Anne Xu, Miles Wingfield, Sarah Stanley, Russell Vance, Aziz Qabar, Astrid, and all members of the Cox lab for their helpful discussions and feedback. This work was supported by the National Institutes of Health grants T32GM132022 (C.M.Q.), U19AI162583 (J.S.C.), U19AI135990 (J.S.C.), 1P01 AI063302 (D.A.P.), 1R01 AI027655 (D.A.P.), as well as the National Science Foundation Graduate Research Fellowship (L.M.W.) and the Henry Wheeler Center for Emerging and Neglected Diseases (C.M.Q.).]

[This work was supported by the National Institutes of Health grants T32GM132022 (C.M.Q.), U19AI162583 (J.S.C.), U19AI135990 (J.S.C.), 1P01 AI063302 (D.A.P.), 1R01 AI027655 (D.A.P.), as well as the National Science Foundation Graduate Research Fellowship (L.M.W., https://nsfgrfp.org/) and the Henry Wheeler Center for Emerging and Neglected Diseases (C.M.Q., https://cend.globalhealth.berkeley.edu). Funders did not play any role in the study design, data collection and analysis, decision to publish, or preparation of the manuscript.]

6. Thank you for stating the following in the Competing Interests section:

[I have read the journal's policy and the authors of this manuscript have the following competing interests: J.D.K. has financial interests in Ansa Biotechnologies, Apertor Pharma, Berkeley Yeast, BioMia, Cyklos Materials, Demetrix, Lygos, Napigen, ResVita Bio, and Zero Acre Farms. D.A.P. has a financial interest in Laguna Biotherapeutics. The other authors declare no competing interests.].

We note that you potentially received funding from a commercial source: [Ansa Biotechnologies, Apertor Pharma, Berkeley Yeast, Cyklos Materials, Demetrix, Lygos, Napigen, ResVita Bio, Zero Acre Farms, and Laguna Biotherapeutics.]

7. We note that you have included the phrase “data not shown” in your manuscript. Unfortunately, this does not meet our data sharing requirements. PLOS does not permit references to inaccessible data. We require that authors provide all relevant data within the paper, Supporting Information files, or in an acceptable, public repository. Please add a citation to support this phrase or upload the data that corresponds with these findings to a stable repository (such as Figshare or Dryad) and provide and URLs, DOIs, or accession numbers that may be used to access these data. Or, if the data are not a core part of the research being presented in your study, we ask that you remove the phrase that refers to these data.

8. Please include your full ethics statement in the ‘Methods’ section of your manuscript file. In your statement, please include the full name of the IRB or ethics committee who approved or waived your study, as well as whether or not you obtained informed written or verbal consent. If consent was waived for your study, please include this information in your statement as well.

9. We note that there is identifying data in the Supporting Information file <Qabar_HMBPP_Supplementary tables>. Due to the inclusion of these potentially identifying data, we have removed this file from your file inventory. Prior to sharing human research participant data, authors should consult with an ethics committee to ensure data are shared in accordance with participant consent and all applicable local laws.

-Location data

Reviewers' comments:

Reviewer's Responses to Questions

**Comments to the Author**

1. Is the manuscript technically sound, and do the data support the conclusions?

Reviewer #1: Partly

Reviewer #2: Partly

Reviewer #3: Partly

2. Has the statistical analysis been performed appropriately and rigorously?

Reviewer #1: Yes

Reviewer #2: No

Reviewer #3: No

3. Have the authors made all data underlying the findings in their manuscript fully available?

Reviewer #1: No

Reviewer #2: Yes

Reviewer #3: Yes

4. Is the manuscript presented in an intelligible fashion and written in standard English?

Reviewer #1: Yes

Reviewer #2: Yes

Reviewer #3: Yes

Reviewer #1: In this paper, Qabar CM et. al. report the generation of a recombinant M. bovis BCG strain that overexpresses an isoprenoid metabolite and phospho-antigen known as HMBPP. HMBPP is a stimulus which expands Vγ9Vδ2 T-cell subsets in the host, and these same cells are believed to play an important role in containing TB disease. The authors postulate that an rBCG strain of this nature would be a more potent TB vaccine than standard BCG.

This concept is innovative and novel. This paper reports a preliminary, partial dataset to support their hypothesis.

Concerns

1. Fig. 3c. There is insufficient evidence to claim that overexpression of GcpE induces Vg9Vd2 cell expansion. The claim is based on Fig. 3c where we are shown on 2 determinations (BCG-WT) compared to 3 determinations (BCG+gcpE-OE). This sample size is too low. In contrast, Fig. 2c (5 dilutions tested on multiple replicates) is convincing evidence that BCG+MEPsynth strain does not elicit the intended Vγ9Vδ2 response. However, we are not shown the same 5 dilutions tested on multiple replicates for the BCG+gcpE-OE strain. The authors should either show more data for the GcpE strain or withdraw the claim that the GcpE overexpressor induces Vg9Vd2 cell expansion.

2. Fig 3c. Why are we only shown the results from Donor E? The legend indicates there was a second donor tested.

3. Since the synthetic biology approach did not increase the expansion of Vγ9Vδ2 T-cells, the authors should consider modifying the title of the manuscript

4. In addition to measuring Vg9Vd2 cell expansion for the recombinant BCGs, it would be valuable to know what the actual levels of HMBPP are.

5. Why were only BCG lysates tested in the Vγ9Vδ2 T-cell expansion assay and not the live BCG strains?

6. Figure 1 has several inconsistencies which should at the very least be discussed as limitations: (i) Why do the mutants show a growth defect in the absence of ATc-mediated knockdown. (ii) Why does the “No guide control” exhibit a growth defect in presence of ATc. (iii) While the spotting assay is a quick and dirty approach, to claim that a gene is essential, a CFU-based growth kinetic analysis should be done. (iv) Lastly, qRT-PCR validation is necessary to show that the knockdowns are functioning correctly.

7. Fig 2b and 3b. Why are mRNA levels so low with these strong promoter constructs at 0.00015 (Fig. 2b) or 0.002 (Fig. 3b) relative to 16S rRNA?

8. Fig. 2c and 2c should have statistics included.

9. Inclusion of the flow cytometry dot plots and gating strategy for Fig. 2c and 3c would be helpful.

10. Line 58-59: The statement that Vγ9+Vδ2+ γδ T cells increase to ~50% of total blood T cells should be qualified. Other studies show lower values closer to 20% (eg Kroca et al. PMID: 10792377)

11. Lines 207-210. Why do the authors think that dxr and ispH appear to be essential in BCG by CRISPRi in their hands, but that in Mtb transposon mutants were viable as reported by the DG Russell lab (ref 63).

12. Line 220. The text states the essential gene rpoB was targeted but Fig 1b shows folC was targeted. Please clarify.

13. Line 329. Discussion should be toned down. Feedback inhibition was a possible reason, but this was not formally demonstrated.

14. The approach for increasing HMBPP used herein was essentially “make more” or “turn up the faucet”. In the discussion, it might be worth mentioning that another way to increase HMBPP levels in BCG would be to conditionally knockdown ispH and thereby reduce usage or “close the drain”

Reviewer #2: The manuscript presents a novel synthetic biology-based strategy to enhance TB vaccine efficacy by engineering the MEP pathway in BCG to increase HMBPP production, thereby stimulating Vγ9Vδ2 T cells. This concept is timely and innovative, combining metabolic engineering with immune modulation to address current limitations in TB vaccination strategies. The rationale and design are well-justified, and the use of human PBMCs enhances translational relevance. However, several critical revisions are required to ensure methodological rigor, data transparency, and appropriate interpretation:

A. Methodological Transparency: Clearly indicate sources of strains, plasmids, antibodies, and include gene identifiers such as Rv numbers or BCG locus tags.

B. CRISPRi Knockdown Assay – Figure 1B

· The spot assay lacks proper controls and normalization:

o Include a dCas9-only or scrambled sgRNA control.

o Ensure equal plating, incubation timing, and inoculum across plates. The no-guide control shows reduced growth upon ATc induction, indicating possible non-specific effects of dCas9 or ATc, which compromises interpretation of gene-specific knockdowns. Additionally, unequal bacterial input and apparent differences in incubation timing between plates further confound comparison. The absence of quantitative measurements (e.g., CFU counts or spot intensity) and replicates limits the robustness of the conclusions.

o Provide CFU counts or quantification of spot intensity across replicates.

· Without these controls, gene essentiality interpretations are weakened.

C. Gene Expression Analysis – Figures 2B and 3B

· The RT-qPCR data should be reanalyzed and presented using the ΔΔCt method.

o Display data as log₂ fold changes with appropriate statistical tests and annotations.

o This will improve clarity, comparability, and rigor.

D. PBMC Stimulation – Figures 2C and 2D

· Normalization of Lysates: OD₆₀₀-based normalization is insufficient:

o Include growth curves or CFU counts to demonstrate equal biomass.

o Specify how lysate input was standardized.

o Authors are requested to quantify HMBPP levels via LC-MS or other methods to support biological interpretation.

· Statistical Annotation (Figure 2D):

o Please include statistical comparisons between:

§ HMBPP vs. BCG WT

§ HMBPP vs. BCG+MEP^synth

o Add p-values and update the legend accordingly.

o Clarify the type and number of replicates used (biological vs. technical) for each experiment.

In particular, for the PBMC-based assays, the number of data points per group appears inconsistent across conditions—for example, in Figure 2D. If the assays were performed using PBMCs derived from the same set of human donors and stimulated simultaneously with different conditions, the number of samples (i.e., data points or “dots” per group) should be identical across all treatment groups. Please clarify the experimental setup and provide an explanation for these discrepancies in sample numbers.

E. Biosafety and Ethical Approvals

· The manuscript must clearly state Institutional Biosafety and IRB approvals for:

o Genetic manipulation.

o Use of human PBMCs.

· Include protocol numbers or justification for exemption, if applicable.

F. Statistical Analysis

· Indicate:

o Statistical tests used in each figure.

o Exact p-values or significance thresholds.

o Confidence intervals where applicable.

G. Discussion

· Expand the discussion on potential limitations, possibility of feedback regulation in MEP pathway, how isoprenoid over-production alters the production of classical antigens of BCG.

· Discuss absence of protective efficacy data. Reframe claims of protective immunity as speculative, pending validation in appropriate in vivo models.

· If you are adding HMBPP quantification, please discuss that.

· Strengthen context with references to:

Clinical efforts involving Vγ9Vδ2 T cell-targeted vaccines.

o Metabolic engineering strategies in mycobacteria.

I. Compliance with Journal Requirements

· Include detailed author contribution statements as per PLOS ONE guidelines.

Overall, the manuscript holds promise but requires substantial revision. Addressing these issues will significantly enhance the robustness and impact of the study.

Reviewer #3: The manuscript presents an innovative synthetic biology approach to enhance BCG-mediated expansion of Vγ9Vδ2 T cells by engineering the isoprenoid biosynthesis pathway in BCG, specifically to overproduce HMBPP. The underlying hypothesis is sound and the objective highly relevant in the context of TB vaccine enhancement strategies, particularly considering the unique role of Vγ9Vδ2 T cells in primate immunity. However, in its current form, the study appears somewhat inconclusive and requires additional data to substantiate its conclusions.

Major Concerns:

1) A critical shortcoming of the study is the lack of direct quantification of HMBPP in the engineered BCG strains. Although gene expression levels of MEP pathway genes and gcpE are provided, they cannot serve as a surrogate for actual metabolite levels. Without intracellular or extracellular quantification of HMBPP, the interpretation that gcpE overexpression leads to increased HMBPP accumulation remains speculative. It is imperative that the authors include HMBPP quantification (e.g., via LC-MS/MS or other validated methods) in both synthetic operon-expressing and gcpE-overexpressing strains to substantiate their claims.

2) In Figures 2C, 2D, and 3C, the number of biological replicates (i.e., PBMC donors) and technical replicates is either unclear or inconsistently described in the legend. This limits the interpretability and statistical robustness of the expansion assay data. Authors should clearly state the number of independent donors, replicate wells, and how variability was accounted for in the analyses (mean ± SD, statistical tests, etc.). A separate sub-section on ‘Statistical Analysis’ should be added to ‘Materials and Methods’ section.

3) While expansion of Vγ9Vδ2 T cells is demonstrated, no functional immune response (e.g., cytokine release, or cytotoxic activity) is assessed to confirm that these expanded cells are immunologically competent. Inclusion of such functional data (IFNγ, Granzyme B, etc.) would strengthen the manuscript and its relevance to vaccine efficacy.

Minor Comments:

1) Reference [29] appears to have a non-standard font or formatting. This should be corrected.

2) The manuscript frequently repeats the background on Vγ9Vδ2 T cells and HMBPP throughout the introduction and results. This dilutes the emphasis on what was experimentally discovered. The authors should consider condensing these sections to improve clarity and focus.

**Do you want your identity to be public for this peer review?** For information about this choice, including consent withdrawal, please see our For information about this choice, including consent withdrawal, please see our Privacy Policy .

Reviewer #1: No

Reviewer #2: **Yes:** Ruchi Jain DeyRuchi Jain Dey

Reviewer #3: No

While revising your submission, please upload your figure files to the Preflight Analysis and Conversion Engine (PACE) digital diagnostic tool, https://pacev2.apexcovantage.com/ . PACE helps ensure that figures meet PLOS requirements. To use PACE, you must first register as a user. Registration is free. Then, login and navigate to the UPLOAD tab, where you will find detailed instructions on how to use the tool. If you encounter any issues or have any questions when using PACE, please email PLOS at . PACE helps ensure that figures meet PLOS requirements. To use PACE, you must first register as a user. Registration is free. Then, login and navigate to the UPLOAD tab, where you will find detailed instructions on how to use the tool. If you encounter any issues or have any questions when using PACE, please email PLOS at figures@plos.org . Please note that Supporting Information files do not need this step.. Please note that Supporting Information files do not need this step.

---

## [Author Response · Author response to Decision Letter 1]

10 Nov 2025

Point-by-point responses to reviewer comments have been uploaded on document upload page.

---

## [Decision Letter · Decision Letter 1]

23 Jan 2026

Dear Dr. Cox,

plosone@plos.org . . A letter that responds to each point raised by the academic editor and reviewer(s). You should upload this letter as a separate file labeled 'Response to Reviewers'.A marked-up copy of your manuscript that highlights changes made to the original version. You should upload this as a separate file labeled 'Revised Manuscript with Track Changes'.An unmarked version of your revised paper without tracked changes. You should upload this as a separate file labeled 'Manuscript'.

We look forward to receiving your revised manuscript.

Kind regards,

Atul Vashist, PhD

Academic Editor

PLOS One

Journal Requirements:

Reviewers' comments:

Reviewer's Responses to Questions

**Comments to the Author**

Reviewer #1: All comments have been addressed

Reviewer #3: All comments have been addressed

2. Is the manuscript technically sound, and do the data support the conclusions?

Reviewer #1: Yes

Reviewer #3: Yes

3. Has the statistical analysis been performed appropriately and rigorously?

Reviewer #1: Yes

Reviewer #3: Yes

4. Have the authors made all data underlying the findings in their manuscript fully available?

Reviewer #1: Yes

Reviewer #3: Yes

5. Is the manuscript presented in an intelligible fashion and written in standard English?

Reviewer #1: Yes

Reviewer #3: Yes

Reviewer #1: All concerns adequately addressed. Crazy that I have to write 100 characters in order for the website to accept this response

Reviewer #3: The revised manuscript shows clear improvement and the work is progressing in a strong direction. One specific point still requires clarification to strengthen the reproducibility of your key findings:

PBMC Expansion Assay Methodology

The PBMC stimulation workflow is appropriate; however, the description of bacterial lysate preparation would benefit from quantitative detail. At present, lysate input is reported only as dilution factors, without specifying the actual concentration of the stimulatory material. Since several processing steps (bead-beating, lysate recovery, 3 kDa filtration, volume adjustments) can alter the final composition, Bacterial culture OD normalization alone may not ensure equivalent stimulant load between samples.

Because the PBMC expansion data (e.g., Fig. 2C, 3C, S2 B–D) are central to the manuscript’s conclusions regarding the differential effects of engineered rBCG versus wild-type BCG, it would substantially strengthen the study to normalize lysates to an absolute metric- such as total protein concentration (μg/mL or μg/well, measured by BCA/Bradford) or a comparable mass-based standard.

For clarity and rigor, it is recommended to include:

Quantification of lysate protein concentration prior to use (μg/mL).

Normalization of all lysates to the same protein input per well during stimulation.

This addition will enhance comparability across experimental arms and ensure that observed differences in PBMC expansion reflect biological effects rather than variation in lysate load. Addressing this point will more firmly support the manuscript’s core conclusions.

**Do you want your identity to be public for this peer review?** For information about this choice, including consent withdrawal, please see our For information about this choice, including consent withdrawal, please see our Privacy Policy .

Reviewer #1: No

Reviewer #3: No

---

## [Author Response · Author response to Decision Letter 2]

4 Feb 2026

We have responded to the reviewer comment in the Response To Reviewers document.

---

## [Decision Letter · Decision Letter 2]

12 Feb 2026

Leveraging a synthetic biology approach to enhance BCG-mediated expansion of Vγ9Vδ2 T cells

PONE-D-25-24963R2

Dear Dr. Cox,

We’re pleased to inform you that your manuscript has been judged scientifically suitable for publication and will be formally accepted for publication once it meets all outstanding technical requirements.

Kind regards,

Atul Vashist, PhD

Academic Editor

PLOS One

Additional Editor Comments (optional):

Reviewers' comments:

Reviewer's Responses to Questions

**Comments to the Author**

Reviewer #3: All comments have been addressed

2. Is the manuscript technically sound, and do the data support the conclusions?

Reviewer #3: Yes

3. Has the statistical analysis been performed appropriately and rigorously?

Reviewer #3: Yes

4. Have the authors made all data underlying the findings in their manuscript fully available?

Reviewer #3: Yes

5. Is the manuscript presented in an intelligible fashion and written in standard English?

Reviewer #3: Yes

Reviewer #3: In the revised manuscript, all concerns adequately addressed. The study represents a step forward in the effort to develop a better vaccine for TB.

**Do you want your identity to be public for this peer review?** For information about this choice, including consent withdrawal, please see our For information about this choice, including consent withdrawal, please see our Privacy Policy .

Reviewer #3: No

---

## [Editor Report · Acceptance letter]

PONE-D-25-24963R2

PLOS One

Dear Dr. Cox,

I'm pleased to inform you that your manuscript has been deemed suitable for publication in PLOS One. Congratulations! Your manuscript is now being handed over to our production team.

Kind regards,

on behalf of

Dr. Atul Vashist

Academic Editor

PLOS One